# Assessing the Concordance of Genomic Alterations between Circulating-Free DNA and Tumour Tissue in Cancer Patients

**DOI:** 10.3390/cancers11121938

**Published:** 2019-12-04

**Authors:** Leila Jahangiri, Tara Hurst

**Affiliations:** 1Department of Life Sciences, Birmingham City University, Birmingham B15 3TN, UK; tara.hurst@bcu.ac.uk; 2Division of Cellular and Molecular Pathology, Department of Pathology, University of Cambridge, Lab blocks level 3, Cambridge Biomedical Campus, Cambridge CB2 0QQ, UK

**Keywords:** solid tumours, circulating-free DNA (cfDNA), somatic genomic alteration (SGA), copy number alterations (CNAs)

## Abstract

Somatic alterations to the genomes of solid tumours, which in some cases represent actionable drivers, provide diagnostic and prognostic insight into these complex diseases. Spatial and longitudinal tracking of somatic genomic alterations (SGAs) in patient tumours has emerged as a new avenue of investigation, not only as a disease monitoring strategy, but also to improve our understanding of heterogeneity and clonal evolution from diagnosis through disease progression. Furthermore, analysis of circulating-free DNA (cfDNA) in the so-called “liquid biopsy” has emerged as a non-invasive method to identify genomic information to inform targeted therapy and may also capture the heterogeneity of the primary and metastatic tumours. Considering the potential of cfDNA analysis as a translational laboratory tool in clinical practice, establishing the extent to which cfDNA represents the SGAs of tumours, particularly actionable driver alterations, becomes a matter of importance, warranting standardisation of methods and practices. Here, we assess the utilisation of cfDNA for molecular profiling of SGAs in tumour tissue across a broad range of solid tumours. Moreover, we examine the underlying factors contributing to discordance of detected SGAs between cfDNA and tumour tissue.

## 1. Introduction

Cancer genomes display a plethora of somatic genomic alterations (SGAs), including single nucleotide variations (SNVs), insertions and deletions (indels), focal amplifications, gene fusions, copy number alterations (CNAs) and numerical and segmental chromosomal alterations (NCAs and SCAs). Chromosome aberrations, including gene fusions and translocations are associated with many tumour types and the vast majority of metastatic cancers harbour chromosome arm level CNAs [1,2]. The current gold standard for molecular profiling in clinical practice is the identification of SGAs, specifically actionable driver alterations in tumour tissue, enabling stratification of patients into precise treatment regimens. In a classical view, driver and passenger alterations differ in actionability and a key task is distinguishing these [3]. A challenge of this task is the heterogeneity among somatic alterations, defined as the emergence of distinct subclones with divergent genetic profiles within the tumour, between the primary tumour and its metastases or between sequential tumour samples [4,5,6]. A potential solution to circumvent this issue, though impractical, is to capture a larger spectrum of the genomic landscape by obtaining multiple biopsies of a primary tumour and its metastases throughout the course of the disease. Alternatively, the use of circulating-free DNA (cfDNA), acquired through the so-called ‘liquid biopsy’, represents a rapid and non-invasive method for genomic profiling. The presence of cfDNA in the plasma or serum of cancer patients and the use of this tool for the analysis of SGAs, including actionable alterations, in cancers of tissues such as lung, breast, gastrointestinal tract, nervous system and prostate has been well documented [7,8,9,10,11,12,13,14,15,16,17,18]. These studies bring into focus the emergence of cfDNA as a potential translational tool for clinical practice, particularly in relation to the analysis of primary and metastatic tumour profiles. Furthermore, cfDNA has emerged as a useful surveillance tool for the early detection and prediction of prognosis in several cancers and has displayed correlations with disease burden and treatment response [19,20,21,22,23,24,25,26,27,28,29]. 

Regarding the potential for the use of cfDNA in routine clinical practice, establishing the extent to which cfDNA reflects the genomic landscape of tumours is significant. However, this task is hampered by numerous technical and biological challenges. Recent studies, though diverse in cohort size and design, have evaluated the feasibility of using cfDNA by measuring the degree of concordance between paired cfDNA and tumour samples and have attempted to dissect the underlying biological or technical factors contributing to discordance. Here we review these studies in the broader context of SGAs in solid tumours. 

## 2. The Concordance Rate of SGAs between cfDNA and Tumour Tissue across Solid Tumours

Mutations in specific oncogenes are frequent signatures in solid and liquid tumours and the presence of these in cfDNA is concordant in variable degrees with that in the tumours. In this section, some studies leveraging the use of cfDNA for detecting SGA in breast, prostate, NSCLC, colorectal, neuroblastoma and oligodendroglioma cancers will be reviewed.

*KRAS* mutations arise in 50% of metastatic colorectal cancer (mCRC) cases, which can affect the response to *EGFR* pathway-targeted therapeutics [30]. In multiple studies of mCRC, cohorts of patients were tested for *RAS* status using standard-of-care PCR and ddPCR (BEAMing) or similar technologies for tissue and cfDNA, yielding 86.4–92% concordance rates [31,32,33]. In excess of 85% of lung cancers are classified as NSCLC, with several actionable alterations of *EGFR* and *ALK* contributing to its pathogenesis [34,35]. In a study conducted by Sung et al., 126 cases of NSCLC patient samples were analysed for concordance of cfDNA and tumour tissue using ultra-deep sequencing and tissue genotyping, respectively. Very high overall concordance rates for *EGFR* mutations (*ex19del* and *L858R*) were observed [15,36,37,38,39,40,41]. 

In the field of breast cancer, circulating tumour cells and cfDNA are promising analytes for prediction of survival and response to therapy [17,20,42,43]. An important cfDNA biomarker of breast cancer, hotspot mutations in *ESR1*, predicts resistance to endocrine therapy [32]. Takeshita et al., compare *ESR1* mutation status of 35 cfDNA and matched tumour tissue in patients with metastatic breast cancer using ddPCR and find an overall concordance rate of 74.3% (26/35) [44]. Further, *PIK3CA* mutations, frequently detected in cfDNA in breast cancer and an indicator of tumour burden and treatment efficacy have been a subject of interest [45,46,47,48] since they also show high concordance between cfDNA and metastatic tumours [49]. 

In a recent study on metastatic prostate cancer, the concordance rate of 45 cfDNA and matched tissue biopsies for clinically-relevant genes was determined by targeted sequencing and whole exome sequencing (WES). This group found copy numbers of clinically actionable genes (i.e., *AR*, *BRCA2*, *PIK3CA*) to be 88.9% concordant between cfDNA and tumour DNA. While rearrangements detected in *PTEN*, *APC*, *BRCA2* displayed 48% concordance [10]. 

In the field of neuroblastoma, *MYCN* amplification status, the strongest indicator of poor prognosis and aggressive behaviour [50,51], has been analysed using cfDNA [16]. In addition to *MYCN*, *ALK* activating alterations occurring in 10% of NB cases have been assayed in cfDNA using PCR-based methods [52]. Combaret et al., used ddPCR to evaluate the mutational status of *ALK* hotspots (*F1174L* exon 23:3520 and 3522, *R1275Q* exon 25:3824) using cfDNA in a cohort of 114 neuroblastoma patients. Their analysis revealed perfect agreement between cfDNA and tumour tissue for the *F1174L ALK* mutation (exon 23:3520), while discordance was observed for the other two mutations [52]. 

A range of SGAs including numerical chromosome alterations (NCAs), segmental chromosome alterations (SCAs) and large SCAs have been investigated in cfDNA in neuroblastoma patients [53,54,55]. Chicard et al., inferred CNAs (including large SCAs, SCAs and NCAs) in cfDNA and matched tumour tissue of neuroblastoma patients. The overall concordance of 97% for dynamic (non-silent) cfDNA and tumour profiles was reported while large SCAs also showed high levels of concordance [53]. In a later publication, this group utilised WES for both cfDNA and tumour tissue and found high concordance of CNAs between cfDNA and primary tumours at diagnosis (151/162) (93%) and with 11/162 (7%) cases of discordance (i.e., 2p gain in tumour only, in case 17) [54]. Good agreement between large structural alterations was also observed by Leary et al., in colorectal and breast cancers. In this study, entire chromosome-level and chromosome arm-level alterations were detected by whole genome sequencing (WGS). Tumour-derived chromosomal copy number changes (1p, 4q loss and 13q gain) and copy number changes of driver alterations including *ERBB2* and *CDK6* were detected in cfDNA of colorectal and breast cancer patients with good concordance rates when tumour tissue was available for analysis [56]. In a study conducted by Lavon et al., statistically significant concordance rates were detected for loss of heterozygosity (LOH) of 10q and 1p (79% and 62%, respectively) between cfDNA and tumour tissue of oligodendroglioma patients [57].

Standing in contrast to the four former studies, Molparia et al., in a cohort of 24 colorectal cancer patients, detect a lack of concordance between CNAs including deletions of 8p,18 and 9p of cfDNA and tumour tissue, highlighting the subclonal nature of CNAs in colorectal cancer [58]. In conclusion, these studies attest to the feasibility of using cfDNA as a tool for detecting a range of SGAs including structural alterations present in most cancers [2,59].

## 3. The Underlying Factors Contributing to Perceived Discordance between SGAs Detected in Solid Tumours and cfDNA

The inter-related technical and biological factors that may contribute to discordance between cfDNA and primary and metastatic tumours will be discussed in detail in this section. Figure 1 shows processing of cfDNA and primary and metastatic tumour tissue from sampling to analysis and the summary of contributing factors to discordance rates observed between SGAs in cfDNA and tumour tissue.

### 3.1. Tumour Fraction and Mutation Allele Frequency (MAF) 

The tumour fraction is the proportion of tumour DNA in total cfDNA. Rapidly proliferating subpopulations of the tumour that outgrow their blood supply prior to apoptosis or necrosis would be expected to release DNA of different sizes into the peripheral blood [60,61]. The location, size and vascularity of the tumour can affect the accessibility of tumour DNA to the circulation, and hence impact tumour fraction [62,63,64]. Therefore, these biological factors can affect the release of tumour DNA in the blood, impacting their representation and detectability in cfDNA [31,32]. For instance, García-Foncillas et al., in a study of metastatic colorectal cancer, report that when SGAs (such as *RAS* mutations) of cfDNA and liver metastasis were compared, a higher concordance rate was obtained than that of cfDNA and lung metastasis [65]. This observation was explained by the higher vascularisation of liver tissue and the greater likelihood of DNA release into the circulation [66]. Furthermore, establishment of the tumour fraction can inform the most appropriate analysis method, especially in the case of alterations presenting at lower MAFs. In a study conducted in a cohort of 520 patients with metastatic prostate or breast cancer, the blood samples of 30% and 40% of breast and prostate cancer patients, respectively, had sufficient tumour fractions for standard depths of WES (i.e., ≥10%) [67]. Regarding MAFs, methods such as ddPCR and NGS with commercial panel designs for molecular profiling, a MAF cut-off value is often introduced [31]. For instance, some ddPCR and NGS platforms have cut-offs in the region of 0.040–.1% and 0.25–%, respectively [68,69], and the commercially-available Guardant360 liquid biopsy assays have almost perfect specificity for SNVs with MAFs of >2% [70]. Therefore, lower concordance is plausible when MAFs fall below the detection cut-off of the method used.

Tumour fraction can be impacted by technical practices used to extract cfDNA from plasma or serum. Guo et al., evaluate the effect of blood sample processing on cfDNA concentration and found that delaying processing beyond four hours significantly decreased detection rate of somatic mutations in cfDNA [71,72]. Release of genomic DNA from white blood cells, resulting in contamination of cfDNA, can be a consequence of delayed processing. The presence of contamination in cfDNA can, however, be accounted for by sequencing white blood cells and filtering somatic mutations attributable to clonal haematopoiesis [73], although this approach will not neutralise the diluting effect of such contamination on the tumour fraction of cfDNA. The necessity of extracting cfDNA from plasma was challenged by a study conducted by Sefrioui et al., The group compared concordance rates in a cohort of 17 mCRC patients and established 93% and 88% mutation detection rates for cfDNA isolated from plasma and crude plasma samples, respectively, suggesting that extraction of cfDNA from plasma may enhance detection by increasing tumour purity [74]. Similarly, increasing cfDNA concentration may also enhance concordance rates, by increasing the tumour fraction [75,76]. For instance, *KRAS* mutation status across 121 patients with NSCLC, melanoma, breast, uterine, pancreatic cancers was compared between cfDNA and matching tumours [77]. In this study, the initial concordance of 85% was improved to 95% by increasing cfDNA concentration in some discordant cases. Finally, tumour fraction can be impacted by cancer type and stage [10]. In a study of multiple cancers, the cfDNA samples of neuroblastoma patients displayed the highest tumour fraction, including two patients in which it was almost 100% [78,79].

Regarding cancer stage, Bettegwada et al., found that circulating tumour DNA, defined as the fraction of tumour DNA in total circulatory free DNA, was detectable in more than 75% of the patients with advanced stage cancers including pancreatic, ovarian, colorectal, melanoma, hepatocellular and head and neck as opposed to 50% of primary brain, renal, prostate or thyroid cancers [19]. In another study conducted by Namløs et al., on gastrointestinal stromal tumours (GIST), patients with metastatic disease displayed significantly higher frequencies of mutation detection in plasma compared to patients with localised disease which correlated with tumour burden. Specifically, all patients (*n* = 10) with metastatic disease had detectable mutations, while this rate was lower for patients in less advanced stages of the disease [80]. Consequently, higher tumour fraction in metastatic diseases may contribute to higher concordance rates observed between cfDNA and metastatic tumour compared to cfDNA and primary tumour [72,81], in addition to indicating poor prognosis [82]. Xie et al., tested 35 pairs of NSCLC primary tumour tissues or metastatic tumours and plasma from treatment-naïve patients using targeted sequencing for a custom panel of 56 lung cancer genes. They interrogated similarities between primary and metastatic tumours and matching cfDNA and observed 62% concordance between the trio (67/108 mutations identified). They also found that the concordance rate improved to 77.3% when they limited their analysis to driver alterations. Interestingly, they observed a higher concordance of cfDNA and metastatic profile (73.2 %) compared to cfDNA and primary tumour profiles (68.4%) [83]. Examples of higher concordance rates in metastatic and cfDNA compared to primary tumour and cfDNA, though not statistically significant, has been reported in Table 1. 

### 3.2. Gene Type and the Effect of Drug Therapy 

In a classical perspective, driver alterations, which are usually truncal and present from early stages of the disease, are SGAs that provide a selective growth advantage. Passenger alterations, which might be neutral or deleterious, are genetically linked to driver alterations [86]. Current views suggest that environmental and treatment variables lead to a more dynamic status of driver versus passenger alterations; for instance, a passenger alteration can transform into a driver alteration [87]. Driver alterations can be deemed “clinically actionable” if an FDA-approved drug or drug under investigation in a clinical trial could target the protein of interest or its downstream effectors [88]. Discordance observed between cfDNA and tumour tissue may be due to the subclonal presentation of drivers in the tumour in later stages of the disease, which can affect the detectability of these drivers in cfDNA and impact concordance rates [86]. A multitude of studies across the field of solid tumours (i.e., prostate, NSCLC, breast, neuroblastoma, renal, gastrointestinal, pancreatic, thyroid and melanoma) focus on individual (Table 2) or multiple malignancies (Table 3) and report a trend for high agreement between cfDNA and tumour tissue with respect to actionable driver alterations, but with notable exceptions [70,89,90,91,92,93,94]. In Table 2 and Table 3, we have included studies that compare the extent to which cfDNA reflects driver and actionable driver alterations of the primary or metastatic tumours. 

Drug therapy may also affect the detection of driver alterations. If a particular clone is resisting therapy through clonal evolution and selection [144,145] and if therapy is stabilising the tumour and suppressing cell turnover, this could potentially affect the representation of clones in the cfDNA pool. Therefore, the selection pressure induced by therapies can lead to heterogeneity within or between tumours, which may, in turn, lead to no or variable representation of subclonal populations within the cfDNA population [5]. In a cohort of 88 NSCLC patients, Schwaederle et al., observed a trend for higher concordance rates of alterations in pre-treatment patients compared to post-treatment patients (64.7% vs. 48.9%) in a subgroup of patients with dynamic (non-flat) cfDNA profiles. Although not statistically significant, differences between the representation of alterations pre- or post-treatment were observed [112]. In a study conducted on *HER2*-positive metastatic breast cancer patients, fluctuations in *HER2* copy number detectable in cfDNA was observed in patients undergoing multiple cycles of therapy. In this cohort of 18 patients, prior to treatment initiation, *HER2* amplification was detectable in cfDNA in only 50% of patients (9/18) (despite *HER2* amplification detection in tumour tissue in all cases (18/18)). In one particular patient, *HER2* copies were not identified in cfDNA prior to treatment and until after cycle 2 of treatment, however, they were detected at an elevated level after cycle 4 of chemotherapy. This level increased further through disease progression after cycle 6 of chemotherapy [146]. This study further supports the effect of cancer treatment on the dynamics of cfDNA release from the tumour.

### 3.3. Sampling and Processing of Tumour Tissue

Due to logistic and safety limitations of taking multiple biopsies in patients [5], detection of SGAs in the peripheral blood has promising potential as a non-invasive alternative [62]. Regarding biopsy sampling intervals, minimising the collection intervals between cfDNA and tumour tissue enhances concordance rates [32,45,147]. Thompson et al., analysed EGFR mutations in advanced NSCLC patients and found that increasing timing intervals between tumour and cfDNA sampling from less than 2 weeks to more than 6 months, led to significantly lower concordance (*p*=0 .038) [15]. In a similar study on a cohort of 88 patients with NSCLC, the overall concordance of *EGFR* mutations varied depending on sampling time, with 88.2% and 64.7% concordance for time intervals of 0.8 months and >0.8 months between blood draw and tissue biopsy, respectively [112]. 

With regards to tumour tissue processing methods, the use of fresh frozen (FF) samples for tumour tissue instead of formalin-fixed paraffin-embedded (FFPE) marginally increased from 57.1% to 66.7%, suggesting fragmentation of DNA in FFPE processing may be significant, especially when the detection assay relies on amplicon-based amplification [72,100,148]. 

### 3.4. Detection Method

Some biomarkers are not easily detectable in the plasma of early stage cancer patients by conventional methods such as ELISA and more sensitive methods such as ddPCR or targeted sequencing may be more appropriate [107,149]. Due to the low abundance of circulating tumour DNA in plasma of early stage cancer patients, using a highly sensitive method while keeping the cost low is challenging. ddPCR assays have high sensitivity and specificity for SNV detection but may not be practical to interrogate a large scope of alterations or unknown alterations, while sequencing approaches including amplicon-based targeted sequencing may profile a broader spectrum of alterations. Demuth and colleagues compared *KRAS* mutation status of 28 patients with metastatic colorectal cancers between cfDNA and matched tumour samples with targeted sequencing and ddPCR, yielding a concordance rate of 79% and 89% for each method, respectively [75]. Similarly, in a study of 127 patients with advanced NSCLC, assaying for driver and drug-resistance alterations, the use of ultra-deep sequencing of cfDNA and orthogonal ddPCR was compared [89]. This study revealed almost identical findings in relation to *EGFR* and *KRAS* mutations by ultra-deep NGS and ddPCR (21/22 cases). In addition, ultra-deep sequencing identified *KRAS* mutations in 17 cases where tumour tissue was deemed insufficient for genotyping, suggesting that ultra-deep targeted sequencing of cfDNA may be instrumental for the identification of specific SGAs in cfDNA missed in tumour tissue. One limitation of targeted sequencing is that analysis is restricted to pre-defined genomic regions, and therefore only patients who display alterations in the analysed regions can be included [150]. 

Depth of sequencing may also play a key role in establishing whether SGAs detected are truncal (ancestral mutation shared by all clones) or subclonal. Chicard et al., in a study of neuroblastoma patients, detected 17 suspected relapse-specific SNVs using WES. However, upon deeper targeted sequencing of the primary tumours, these SNVs were identified in minor subclones present at diagnosis [54]. 

Sequencing depth is also significant in the case of detection of CNAs including chromosome-level copy number or structural changes and rearrangements [56]. In a study of multiple paediatric solid tumours, Klega et al., use ultra-low-pass WGS (ULP-WGS) with a coverage of 0.2x to 1x for detecting CNAs in cfDNA [78]. Another study demonstrated that WGS with a shallow coverage of 0.1x is sufficient for reliable analysis of CNAs [151]. This method was also successfully leveraged in metastatic prostate cancer [18]. This study revealed chromosome arm gains and losses, high level copy number gains, fusions and SNVs indicated in the pathogenesis of prostate cancer, therefore the timely and costly deep coverage WGS may be avoidable. Finally, the use of either the same analysis platform for both cfDNA and tumour or analysing all samples on both platforms may reduce the discordance rates attributed to differences in sensitivity [55]. 

### 3.5. Heterogeneity

Subsequent to fine-tuning of methods and practices for detecting technical and biological artefacts in cfDNA and tumour tissue comparative studies, the degree of contribution of tumour heterogeneity and clonal evolution to differences between matched cfDNA and tumour biopsies can be evaluated. 

In a study on renal cell carcinoma, it was shown that 65% of SGAs were not detectable in every region of the primary and metastatic tumours. In addition, intratumoural heterogeneity was observed in relation to specific tumour suppressor genes [5]. These results suggest the presence of subclones, within the primary tumour that may compete or collaborate. These subclones may both evolve and expand through disease progression, leading to divergence of genomic landscapes [152], in addition to increasing adaptability to the dynamic microenvironment of the tumour [4].

The presence of spatial and temporal tumour heterogeneity has been detected in studies comparing cfDNA alterations with primary or metastatic tumours. A good example of this is a study conducted in gastroesophageal adenocarcinoma comparing the genomic profiling of primary and metastatic lesions by sequencing across multiple cohorts. This study found extensive differences in SGAs including actionable alterations between primary and metastatic tumours. One key observation was the high concordance rate of 87.5% for actionable alterations between cfDNA and metastatic tissue that were originally found to be discordant between primary and metastatic tumours. This valuable observation may implicate cfDNA in providing a representation of malignant disease, in addition to highlighting heterogeneity between alterations of primary and metastatic tumours [21,153,154], although high agreement between primary and metastatic tumour SGAs has been reported in other studies [155,156]. 

## 4. The future of cfDNA in precision oncology

Precision oncology applies tailored treatment to individual characteristics of patients by detecting and monitoring actionable alterations to inform targeted therapy and patient management strategies. The tumour biopsy remains the most efficient diagnostic tool at present but due to the impracticality of obtaining multiple tumour samples to capture a larger scope of spatial and temporal heterogeneity [157], the use of cfDNA has emerged as a viable alternative [158]. The emergence of cfDNA as a clinically relevant, minimally invasive tool to inform disease burden, acquisition of actionable alterations and resistance to therapy has been extensively documented [20,28,42]. Despite these promising prospects, the extent to which cfDNA captures and reflects the SGAs of the tumour and its metastases is not fully dissected. Furthermore, cfDNA as a diagnostic tool poses limitations; for instance, cfDNA cannot replace histologic information obtained from tissue biopsy and the dynamics of cfDNA release may lead to variable representation of important actionable alterations in the cfDNA population [60,61,62,63,64]. Considering these limitations, many studies suggest a “companion role” for cfDNA in the clinical diagnostic setting, in which tumour tissue and cfDNA samples could be considered in parallel to improve the likelihood of early detection of actionable alterations in patients. This approach would provide a window of opportunity for early detection and initiation of targeted therapy, especially in cases where actionable alterations are detected in cfDNA and not tumour tissue due to sampling bias inherent in tissue biopsies [15,36,54,73,89,159]. When tumour biopsies cannot be obtained safely or tumour tissue is not available, plasma sampling could provide valuable information for clinical decision-making.

In addition to the utility of cfDNA as a diagnostic tool, clinical resistance to therapy can be longitudinally monitored in time by analysing serial cfDNA samples [20,160]. Resistance to therapy can emerge from the acquisition of SGAs in genes and pathways targeted by therapy. Understanding the mechanism of resistance by analysis of tumour tissue is challenging due to safety issues of serial tissue biopsies. Hence, the serial mutation profiling based on cfDNA over the duration of the diseas, may permit the real-time appreciation of the efficacy of systemic therapy and detection of disease resistance. In multiple studies, cfDNA has been utilised for monitoring of resistance mutations including *EGFR T790M*, *BRAF*, *ALK*, *ERBB2* amplification in NSCLC patients receiving therapy [15,37,41,73,109,128,161,162]. In a study by Sung et al., longitudinal cfDNA analysis lead to the detection of *EGFR T790M* mutation emergence in 28.6% of NSCLC patients receiving *EGFR* TKI treatment [36]. Also longitudinal cfDNA analysis has been used in colorectal cancer patients for monitoring of cetuximab resistance through acquisition of secondary *KRAS* mutations [139] and also resistance to antiangiogenic therapies [163]. 

## 5. Conclusions

In conclusion, the importance of this study was the review of the feasibility of using cfDNA for detecting a range of SGAs of tumours across the broad field of solid tumours. Further, concordance rates of actionable driver alterations across solid tumours were examined. Despite some inconsistencies, a trend of high concordance rates for these alterations detected in plasma and tumour tissue was observed. However, due to factors such as clonality and treatment that may affect these rates, we would suggest for each study to be considered in the specific context of cycle of treatment, method, cancer type and stage. Actionable alteration status is critical to targeted therapy decision-making and monitoring treatment response and the promising prospect of leveraging plasma cfDNA for detecting and monitoring these alterations is clinically relevant. In addition, this study drew examples from the literature to interrogate the technical challenges that impact agreement rates between tumours and cfDNA. Fine-tuning of methods and practices is warranted to confidently dissect and distinguish heterogeneity from artefacts. As more sensitive and affordable sequencing technologies become available, deep sequencing of cfDNA can provide insight into tumour evolution and monitoring treatment resistance in several cancers [159].

## Figures and Tables

**Figure 1 cancers-11-01938-f001:**
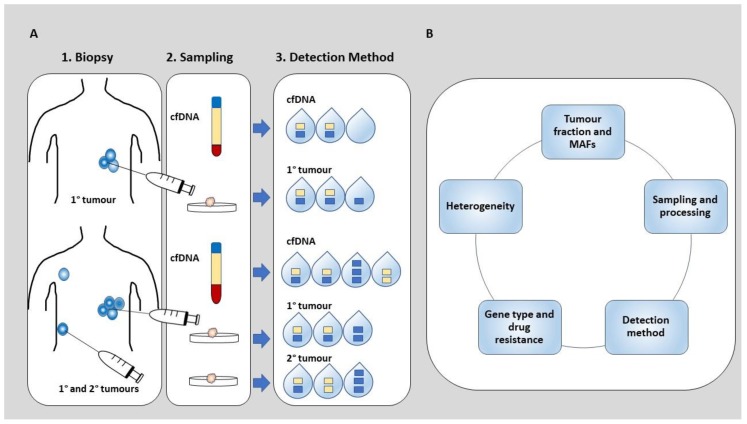
Analysis of cfDNA and tumour tissue in patients with solid tumours: (A) tissue biopsies and cfDNA samples of patients with primary or metastatic tumours are processed for detection of specific somatic genomic alterations (SGAs). The concordance rate between cfDNA and primary tumour or cfDNA and primary and metastatic tumours can be evaluated (B) A summary of factors affecting concordance rates between cfDNA and tumour tissue, 1^o^: primary tumour, 2^o^: secondary (metastatic) tumour.

**Table 1 cancers-11-01938-t001:** Example summary data of trends for high concordance rates between somatic GAs of cfDNA and metastatic tumours compared to cfDNA and primary tumour.

Author/Cohort Size	Cancer Type	Concordance with Primary or Metastatic Tumour	Driver and Actionable Driver Alterations	Method for Tumour/cfDNA
Thompson/102 [15]	NSCLC	cfDNA and primary tumour (51%) compared to cfDNA and metastatic tumour (79%) for all alterations	50 drivers and 12 resistance alterations	Targeted sequencing
Liu/72 [81]	NSCLC	cfDNA and primary (50%) compared to cfDNA and metastatic (65%) in 19 patients	lung cancer panel including *EGFR L858R,* *L861Q,* *e19 del, e20INS, KRAS G12X, EML4-ALK, RET-KIF5B, BRAF V600E*	ARMS-PCR and targeted sequencing/Sequencing and ddPCR
Xie/35 [83]	NSCLC	cfDNA and metastatic tumour (73.2 %), cfDNA and primary tumour (68.4%)	56 lung cancer genes	Targeted sequencing
Guo/56 [72]	NSCLC	54.6% of patients in early stage and 80% in late stage	lung & colon cancer panel (LV103) and lung cancer panel (L82)	Targeted sequencing for both, ddPCR for some cfDNA samples
Garcia- Saenz/49 [47]	6 metastatic and 43 localised breast cancer	59.1% (overall) 79.8% (for metastatic patients)	*PIK3CA* mutations	COBAS *PIK3CA* Mutation Test/ddPCR using (rare *PIK3CA* Mutation Assays)
Tzanikou/56 [84]	Early and metastatic breast cancer	48.2% (27/56) in early breast cancer, 66.6% (18/27) in metastatic breast cancer	*PIK3CA* mutations	Custom method and ddPCR
Chae/12 [70]	mCRC	For sequencing approaches, 39% for primary and 55% for metastasis in all panel	21 gene panel including *TP53, PIK3CA* and *KRAS*	Targeted sequencing/targeted sequencing, OnTarget assay and ddPCR
Kato/55 [85]	Esophageal, gastroesophageal junction, and gastric adenocarcinoma	concordance between ctDNA and primary site vs. cfDNA and metastatic site for *TP53*: 52.2% vs. 87.5% and for *ERBB2*: 78.3% vs. 100%	54-73 gene panel including *KRAS, TP53* and *PTEN*	Sequencing

NSCLC: non-small cell lung cancer; mCRC: metastatic colorectal cancer. Studies using cfDNA that investigated (1) the mechanism of resistance to drug therapy and (2) the comparison of methods and (3) efficacy of drug therapy have not been included, but studies that compare the extent of which cfDNA and reflect driver and actionable driver alterations of the tumours, primary and metastatic have been included.

**Table 2 cancers-11-01938-t002:** Comprehensive summary data for driver and actionable driver alterations concordance rates between cfDNA and tumours in individual cancer types.

Author/Cohort Size	Cancer Type	Concordance Information	Positive Concordance ^(MUT/MUT)^|Negative Concordance ^(WT/WT)|^Discordance	Driver and Actionable Driver Alterations	Method for Tumour/cfDNA
Wyatt/45 [10]	MPC	88.9% in clinically actionable genes		72 genes including *AR, BRCA2, PTEN, PIK3CA* and *TP53*	WES/targeted sequencing
Vandekerkhove/53 [95]	MPC	80% in matched samples		Panel of genes including *TP53* and DNA repair genes	Targeted sequencing
Grasselli/146 [31]	mCRC	89.7%	10.3% (15 cases) concordance	*RAS* mutations	SoC PCR/ddPCR (BEAMing)
Bando/280 [32]	mCRC	86.4% (242/280)	82.1% (110/134)|90.4% (132/146)|11% (38/280)	*RAS* mutations	ddPCR (BEAMing)
Garcia-Foncillas/236 [65]	mCRC	89% (210/236) improved to 92% by re-analysis	86.30%|92.40%|In lung metastasis cases (tissue only)	*RAS* mutations	SoC PCR/OncoBEAM
Schmiegel/98 [33]	mCRC	91.8% (90/98)	90.4% (47/52)|93.5% (43/46)|-	*RAS* mutations	Sequencing, SOC, ddPCR (BEAMing)/ddPCR (BEAMing)
Demuth/28 [75]	mCRC	79% for Ion Torrent seq.- 89% for ddPCR		*KRAS* mutations	Genotyping/Sequencing and ddPCR
Spindler/229 [96]	mCRC	85%		*KRAS*	Standard methods/ARMS-qPCR
Bachet/425 [97]	mCRC	71%- 89%		*RAS*	Standard methods/sequencing
Vidal/115 [98]	mCRC	93%		*RAS*	Standard methods/OncoBEAM
Buim/26 [99]	mCRC	71%		*KRAS*	Standard methods/pyrosequencing
Thierry/140 [66]	mCRC	72%, 74% and 87% for *KRAS* exon 2, *KRAS* exon 3–4 and *BRAF V600E,* respectively		28 mutations including *KRAS, BRAF, NRAS*	Standard methods/Q-PCR-based-method (IntPlex V)
Wang/184 [100]	mCRC	93.33% in pre-treatment cohort		*KRAS, NRAS, BRAF, PIK3CA*	ARMS-based PCR /Firefly
Osumi/101 [101]	mCRC	77.2% (78/101) for *RAS*	23 cases for *RAS* (discordance)	14 CRC- related genes including, *APC, TP53* and *RAS*	Standard methods/Sequencing
Germano/20 [102]	mCRC	84.6% (11/13 cases)		*RAS, BRAF, ERBB2*	Standard methods/ddPCR
Beije/12 [103]	mCRC	*KRAS, PIK3CA* and *TP53* for OnTarget assay (80%), digital PCR (93%)		21 CRC gene panel including *TP53, PIK3CA* and *KRAS*	Sequencing/Sequencing, OnTarget assay and ddPCR
Kato/94 [104]	CRC	ranging from 63.2% *APC* to 85.5% *BRAF*		panel including *KRAS, TP53* and *APC*	Sequencing
Mohamed Suhaimi/44 [105]	CRC	84.1% for *KRAS* and 90.9% *BRAF*		*KRAS* and *BRAF*	Genotyping/sanger sequencing, HRM and ASPCR and pyroseqeuncing
Takeshita/35 [44]	MBC	74.3% (26/35)	1/35|25/35|9/35	*ESR1* mutations	ddPCR
Beaver/29 [48]	Early BC	14/15 mutations		*PIK3CA* mutations	Sanger sequencing, ddPCR/ddPCR
Higgins/49 and 60 [45]	MBC (49 retrospective and 60 prospective)	100% in 41 matched retrospectives, 72.5% in 51 prospectives	27.5% in 51 prospective samples (discordance)	*PIK3CA* mutations	Sequencing or BEAMing/ddPCR (BEAMing)
Chae/45 [70]	BC	91.0%–94.2% for all genes	10.8%–15.1% (3.5% for CNAs) positive concordance		Foundation 1/Guardant360
Board/76 [46]	46 metastatic, 30 localised BC	95% in 41 matched samples	80%|(47%) discordance	*PIK3CA* mutations	Standard methods/ARMS PCR*
Garcia- Saenz/49 [47]	6 Metastatic and 43 localised BC	59.1% (overall) 79.8% (for metastatic patients)		*PIK3CA* mutations	COBAS *PIK3CA* Mutation Test/ddPCR using (rare *PIK3CA* Mutation Assays)
Kodahl/66 [49]	*PIK3CA*- mutated MBC	83% (20/24 cases)		*PIK3CA* mutations	ddPCR
Combaret/114 [52]	NB	100%	1/1|1/1|0	*ALK; F1174L* (e23: 3520, T>C)	ddPCR and targeted sequencing
		55 cases	6 cases|49 cases|4 (cfDNA only), 1 (tumour only)	*ALK, F1174L* (e23:3522, C>A)	
		58 cases	12 cases|46 cases|1 (cfDNA only), 1 (tumour only)	*ALK; R1275Q* (e25:3824, G>A)	
Kurihara/10 [106]	NB	100%	2/2|8/8|0	*MYCN*	FISH/ddPCR
Chen/58 [107]	Stage IA, IB, and IIA NSCLC	50.4%		Panel of 50 driver alterations including *EGFR, KRAS, PIK3CA* and *TP53*	Targeted sequencing
Sung/126 [36]	NSCLC	90% (*ex19del*), and 88.33% (*L858R*)		*EGFR* (*ex19del* and *L858R*)	Genotyping/Targeted sequencing and ddPCR
Li/164 [108]	NSCLC	73.6%		*EGFR* mutations	ARMS
Lee/81 [37]	NSCLC	86.2% (*ex19del*) and 87.9% (*L858R*)		*EGFR* (*ex19del* and *L858R*)	Genotyping/ddPCR
Thompson/102 [15]	NSCLC	79% (19/24) for actionable *EGFR* mutations 97.5% across all variants	60% across all variants	50 drivers, 12 resistance alterations	Sequencing
Jin/69 [109]	NSCLC	88.2% for *EGFR* mutations		*EGFR Ex19del, L858R, G719S/C*, and *L861Q,* *TP53 mutations, amp. of RB1, PIK3CA and MYC*	Targeted Sequencing
Yang/73 [68]	NSCLC	74% (54/73)	26% (19/73) (discordance)	*EGFR* mutations	Sequencing/Sequencing and ddPCR
Guo/41 [110]	NSCLC	78.1%		50 cancer genes including *EGFR, KRAS*, and *TP53*	Targeted sequencing
Villaflor/68 [111]	NSCLC	High concordance for truncal oncogenic drivers, 71% for *EGFR*		Driver alterations including *EGFR*	targeted multiplex testing or tissue- based sequencing/Guardant360
Liu/72 [81]	NSCLC	54.2% for all clinically actionable alterations, *EGFR L858R* (93.1%), *EGFR e19 del* (90.3%), *KRAS G12X* (96.9%), *ALK rearrang.* (96.9%)	*MET* or *HER2* CNA in cfDNA but not tumour (discordance)	*EGFR L858R,* *L861Q,* *e19 del, e20 INS, KRAS G12X, EML4-ALK, RET-KIF5B and BRAF V600E*	ARMS-PCR and sequencing/Sequencing (cfDNA also validated by ddPCR)
Schwaederle/88 [112]	NSCLC	76.5- 80.8 % for *EGFR* mutations depending on sampling time	7/26 (*EGFR* mutations) 53% for all alterations|14/26 (*EGFR* mutations)|5/26 (*EGFR* mutations) 2 cfDNA only, 3 tumour only	Mutations in *TP53, EGFR, MET, KRAS* and *ALK*	Sequencing or genotyping or no test/Guardant360
Yang/107 [113]	NSCLC	74.8% (80/107) *EGFR* 88.8% (95/107) *BRAF*		*EGFR* and *BRAF* mutations	Standard methods/competitive Allele-Specific TaqMan PCR (CastPCR)
Soria- Comes/102 [114]	NSCLC	87.4%		*EGFR* mutations	Cobas *EGFR* assay
Yu/22 [115]	Advanced NSCLC	For *19DEL* and *L858R* (90% and 95%, respectively)		*EGFR* mutations (*19DEL* and *L858R)*	ARMS/ddPCR
Mok/241 [116]	Advanced NSCLC	88% (209/238)		*EGFR* mutations	Cobas 4800 FFPET test/Cobas 4800 blood test
Zhu/51 [117]	Advanced NSCLC	86.73%		*EGFR* mutations	Standard methods/ddPCR
Yao/39 [118]	Advanced NSCLC	78.21% (30.5/39) for all genes	47.43%|30.77%|21.8%	Panel of 40 genes including *EGFR, KRAS, PIK3CA, ALK* and *RET*	Targeted sequencing
Cui/180 [119]	Advanced NSCLC	87.8%	97.3%|85.3%	*EGFR* mutations	Standard methods/SuperARMS
Leighl/282 [120]	Advanced NSCLC	98.2% for *EGFR, ALK, ROS1, BRAF*			SoC PCR/Guardant360
Wu/50 [121]	Advanced NSCLC	86% (43/50 cases)		Driver alterations including *EGFR, TP53, RB1*	Sequencing
Sim/50 [122]	Advanced NSCLC	81% for *EGFR*		*BRAF, EGFR, ERBB2, KRAS, NRAS, PIK3CA*	Sequencing
Xu/42 [123]	Advanced NSCLC	Overall 76%		*EGFR, KRAS, PIK3CA*, and *TP53*	Targeted sequencing
Reck/1311 [124]	Advanced NSCLC	89% (in 1162 matched samples)		*EGFR* mutations	Standard methods of local centres
Jia/150 [125]	Advanced NSCLC	94.7% for *EGFR* and *RAS*		*EGFR* and *KRAS* mutations	Standard methods/ddPCR
Veldore/132 [126]	Advanced NSCLC	96.96%		*EGFR* mutations	Standard methods/sequencing
Ma/219 [127]	Advanced NSCLC	82%		*EGFR* mutations	ARMS
Denis/1311 [128]	Advanced NSCLC	96% in 126 matched samples		*EGFR* mutations	Standard methods
Guibert/46 [129]	Advanced NSCLC	*ROS1/ALK* (8/9), *EGFR* (9/9), *BRAF/MET/HER2* (4/6)		*EGFR* mutations*, ROS1, ALK, BRAF/MET/HER2*	Standard methods/Sequencing and ddPCR
Hahn/19 [90]	mRCC	8.6% concordance	DNA repair genes (discordance)		Foundation 1/Guardant360
Howell/51 [130]	HCC	moderate		*ARID1A AXIN1, ATM, CTNNB1, HNF1A* and *TP53*	Targeted sequencing
Bernard/194 [131]	PDAC (localised or metastatic)	>95% for *KRAS* in surgically resected tissue		*KRAS*	ddPCR
Cohen/221 [93]	PDAC	100%		*KRAS* mutations	Sequencing
Pishvaian/34 [94]	Pancreatic cancer	Low concordance		Panels including *KRAS* and *TP53*	Foundation 1/Guardant360
Kinugasa/75 [132]	Pancreatic cancer	77.3% (58/75)		*KRAS*	PCR-PHFA/ddPCR
Gangadhar/25 [133]	Advanced melanoma	81.8% (9/11)		61 gene panel including *BRAF, NRAS* and *KIT*	Standard methods/Sequencing
Haselmann/634 [134]	Melanoma	*BRAFV600* (92.3%–94.5%)		*BRAF*	SoC PCR/BEAMing
Tang/58 [135]	Melanoma	70.2%		*BRAF*	Standard methods/3D ddPCR
Pinzani/55 [136]	Melanoma	80%		*BRAF*	Allele-specific RT-PCR
Calapre/24 [137]	Advanced melanoma	80% (in a subgroup of 7 matching tissue and cfDNA)		30 melanoma genes including *BRAF, NRAS, NF1* and *TERT*	Targeted sequencing (ddPCR for some cfDNA cases)
Sandulache/23 [138]	Anaplastic thyroid carcinoma	high for *BRAF, PIK3CA, NRAS,* and *PTEN* and moderate for *TP53*	Highest discordance in post-treatment patients	50 gene panel for tissue, 70 gene panel for cfDNA, including *BRAF, NRAS, TP53* and *PIK3CA*	Sequencing

MPC: metastatic prostate cancer; mCRC: metastatic colorectal cancer; MBC: metastatic breast cancer; NB: neuroblastoma; NSCLC: non- small cell lung cancer; mRCC: metastatic renal cell carcinoma; HCC: hepatocellular carcinoma; PDAC: pancreatic ductal adenocarcinoma. ARMS: amplification refractory mutation system with scorpion probes. Positive concordance refers to mutant cfDNA/mutant tumour tissue, whereas negative concordance refers to WT cfDNA/WT tumour tissue. Overall concordance includes positive and negative concordance.

**Table 3 cancers-11-01938-t003:** Comprehensive summary data for driver and actionable driver alterations concordance rates between cfDNA and tumours in studies of multiple cancer types.

Author/Cohort Size	Cancer Type	Concordance or Discordance Information	Driver and Actionable Driver Alterations	Method for Tumour/cfDNA
Kim/75 [139]	CRC, melanoma gastrointestinal stromal tumour, renal cell carcinoma, gastric cancer, sarcoma and 4 other cancers	85.9% when all detected mutations considered across all tumour types	Panel of 54 cancer genes	Sequencing
Rachiglio/79 [140]	44 metastatic NSCLC and 35 mCRC	High concordance for *EGFR* (17/22) and lower concordance for other drivers	*ALK, EGFR, ERBB2, ERBB4, FGFR1, FGFR2, FGFR3, MET, DDR2, KRAS, PIK3CA, BRAF, AKT1, PTEN, NRAS, MAP2K1, STK11, NOTCH1, CTNNB1, SMAD4, FBXW7, TP53*	Sequencing/Sequencing and ddPCR
Phallen/200 [141]	Breast, colorectal, Lung, Ovarian cancer	High concordance	58 cancer related genes including drivers	Sequencing (TEC-Seq)
Riviere/213 [9]	colorectal adenocarcinoma, appendiceal adenocarcinoma, hepatocellular carcinoma, pancreatic ductal adenocarcinoma	96% *KRAS* amplification, 94% *MYC* amplification, 95% *KRAS G12V*, 91% *EGFR* amplification 96% overall concordance on gene level	Panel of 68 genes including *KRAS* amplification, *MYC* amplification, *KRAS G12V, EGFR* amplification	Guardant360 panel
Jovelet/334 [76]	thoracic, gastrointestinal, breast, head and neck, gynaecologic and urologic cancers	On a gene level only 173/347 mutations corresponded between cfDNA and tumour tissue, 174/347 discordant mutations	Panel of 50 cancer hotspots V2 (CHP2) including *TP53, KRAS, PIK3CA, EGFR, APC*	Sequencing
Leary/91 [56]	Colorectal or breast cancer	Good concordance for cancer driver genes such as *ERBB2* and *CDK6*	Chromosomal alterations including rearrangements of *CDK6* and *ERBB2* loci	Sequencing
Toor/28 [89]	advanced stage gastrointestinal and lung malignancies	7% for lung subgroup, 8% for gastrointestinal subgroup (90% positive concordance), high discordance with respect to driver and actionable alteration		Caris or paradigm panels/Guardant360 panel
Baumgartner/80 [142]	appendix cancer, colorectal, peritoneal mesothelioma, small bowel, cholangiocarcinoma, ovarian, and testicular cancer	Overall, positive, and negative concordance was 96.7%, 35.3%, and 96.6% (in 15 cases with matched samples)	Panel of genes including *TP53* and *KRAS*	Sequencing
Kato/55 [85]	Esophageal, gastroesophageal junction, and gastric adenocarcinoma	61.3% (*TP53* alterations) to 87.1% (*KRAS* alterations)	54-73 gene panel Including *KRAS, TP53* and *PTEN*	Sequencing
Perkins/105 [143]	Colorectal, melanoma, breast, prostate, ovarian, NSCLC, mesothelioma, sarcoma, glioblastoma, ACUP, cholangiocarcinoma, and cervical, endometrial, duodenal, esophageal, pancreatic and renal cancers	Overall 60% (25/42)	*BRAF, KRAS, NRAS, HRAS, MET, AKT, PIK3CA, KIT*	Standard methods/Mass Spectrometry TypePLEX and OncoCarta panel (v1.0)

CRC: colorectal cancer; mCRC: mCRC: metastatic colorectal cancer; NSCLC: non-small cell lung cancer; ACUP: adenocarcinoma of unknown primary.

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
