# Peer review of "Assessing the Concordance of Genomic Alterations between Circulating-Free DNA and Tumour Tissue in Cancer Patients"

_cancers, 2019, doi:10.3390/cancers11121938_

Round 1
Reviewer 1 Report
A very comprehensive review on the concordance of genomic alterations between CFS DNA and tumour tissue in cancer patients. The authors have put together a nice presentation with comments and suggestions about the potential use of CFS DNA as a tool for molecular profiling of somatic genomic alterations in some solid tumours. As this is a review article the authors presented results from previous published data and they were right to write about the technical challenges involved in using such an approach. It would be nice to mention whether CFS DNA could be applicable to other solid tumours i.e. brain tumours.
Author Response
A very comprehensive review on the concordance of genomic alterations between CFS DNA and tumour tissue in cancer patients. The authors have put together a nice presentation with comments and suggestions about the potential use of CFS DNA as a tool for molecular profiling of somatic genomic alterations in some solid tumours. As this is a review article the authors presented results from previous published data and they were right to write about the technical challenges involved in using such an approach. It would be nice to mention whether CFS DNA could be applicable to other solid tumours i.e. brain tumours.
We sincerely thank the reviewer for his/ her comments. In regards to the question asked, we would like to cite the following papers, in which the feasibility of cfDNA for detecting epigenetic, miRNA and genetic markers of glioblastomas and oligodendromas have been evaluated:
doi: 10.1093/neuonc/nop041
doi: 10.1371/journal.pone.0184969
The paper from Lavon et al (doi: 10.1093/neuonc/nop041) has now been incorporated into the review (lines 107-110). The other paper (doi: 10.1371/journal.pone.0184969) considers epigenetic markers that is beyond the scope of our review. We hope the respectable reviewer finds this change satisfactory.

Reviewer 2 Report
The authors review the performance of liquid biopsies in detecting actionable somatic genomic alterations as compared to direct biopsies. The authors clearly explain its need and identify good concordance between the two methods, as well as reasonable technical and biological explanations for cases of discordance. I commend the authors on their efforts and offer the following comments:
-In section 2, the authors have chosen “The feasibility of detecting SGAs in cfDNA across solid tumors” as its title. “The concordance rates …” may be more appropriate since technical feasibility was not discussed.
-Figure 1 is unclear and needs more labels, as well as more detail in the legend and either a graphical link between the left and right halves or their separation into distinct panels.
-240-242: "the use of fresh frozen (FF) samples for 240 tumour tissue instead of formalin-fixed paraffin-embedded (FFPE) marginally reduced from 66.7% 241 to 57.1%, suggesting fragmentation of DNA in FFPE processing may be significant,". I think the authors intended to say that FFPE dropped it from 66 to 57%, not FF?
Author Response
The authors review the performance of liquid biopsies in detecting actionable somatic genomic alterations as compared to direct biopsies. The authors clearly explain its need and identify good concordance between the two methods, as well as reasonable technical and biological explanations for cases of discordance. I commend the authors on their efforts and offer the following comments:
We sincerely thank the reviewer for his/her comments.
-In section 2, the authors have chosen “The feasibility of detecting SGAs in cfDNA across solid tumors” as its title. “The concordance rates …” may be more appropriate since technical feasibility was not discussed.
We thank the reviewer for this comment and we have changed the title of the section to: “The concordance rate of SGAs between cfDNA and tumour tissue across solid tumours”. We hope the respectable reviewer finds this change satisfactory.
-Figure 1 is unclear and needs more labels, as well as more detail in the legend and either a graphical link between the left and right halves or their separation into distinct panels.
We thank the reviewer for his/her comment. We have added labels to the figure and provided more detail in the figure legend and the figure itself. In addition we have separated the left and right halves into distinct panels using appropriate annotation (A and B). We hope the respectable reviewer finds this change satisfactory.
-240-242: "the use of fresh frozen (FF) samples for 240 tumour tissue instead of formalin-fixed paraffin-embedded (FFPE) marginally reduced from 66.7% 241 to 57.1%, suggesting fragmentation of DNA in FFPE processing may be significant,". I think the authors intended to say that FFPE dropped it from 66 to 57%, not FF?
We thank the reviewer for his/her comment and apologise for the oversight. We have changed the sentence to: With regards to tumour tissue processing methods, the use of fresh frozen (FF) samples for tumour tissue instead of formalin-fixed paraffin-embedded (FFPE) marginally increased from 57.1% to 66.7%, suggesting fragmentation of DNA in FFPE processing may be significant, especially when the detection assay relies on amplicon-based amplification [71,96,97]. We hope the respectable reviewer finds this change satisfactory.
